# Remediation of Soil Polluted with Cd in a Postmining Area Using Thiourea-Modified Biochar

**DOI:** 10.3390/ijerph17207654

**Published:** 2020-10-20

**Authors:** Yanfeng Zhu, Jing Ma, Fu Chen, Ruilian Yu, Gongren Hu, Shaoliang Zhang

**Affiliations:** 1Engineering Research Center of Ministry of Education for Mine Ecological Restoration, China University of Mining and Technology, Xuzhou 221008, Jiangsu, China; 18014087032@stu.hqu.edu.cn (Y.Z.); jingma2013@cumt.edu.cn (J.M.); slzhang@cumt.edu.cn (S.Z.); 2College of Chemical Engineering, Huaqiao University, Xiamen 361021, China; ruiliany@hqu.edu.cn; 3Low Carbon Energy Institute, China University of Mining and Technology, Xuzhou 221008, Jiangsu, China

**Keywords:** biochar, thiourea, cadmium, soil, immobilization

## Abstract

Cadmium presence in soil is considered a significant threat to human health. Biochar is recognized as an effective method to immobilize Cd ions in different soils. However, obtaining effective and viable biochar to remove elevated Cd from postmining soil remains a challenge. More modifiers need to be explored to improve biochar remediation capacity. In this investigation, pot experiments were conducted to study the effects of poplar-bark biochar (PBC600) and thiourea-modified poplar-bark biochar (TPBC600) on Cd speciation and availability, as well as on soil properties. Our results showed that the addition of biochar had a significant influence on soil properties. In the presence of TPBC600, the acid-soluble and reducible Cd fractions were transformed into oxidizable and residual Cd fractions. This process effectively reduced Cd bioavailability in the soil system. Compared to PBC600, TPBC600 was more effective in improving soil pH, electrical conductivity (EC), organic matter (SOM), total nitrogen (TN), ammonium nitrogen (${\mathrm{NH}}_{4}^{+}-N$), nitrate nitrogen (${\mathrm{NO}}_{3}^{-}-N$), available potassium (AK), available phosphorus (AP), and available sulfur (AS). However, this improvement diminished as incubation time increased. Results of Pearson correlation analysis, multivariate linear regression analysis, and principal component analysis showed that soil pH and available phosphorus played key roles in reducing the available cadmium in soil. Therefore, TPBC600 was shown to be an effective modifier that could be used in the remediation of soil polluted with Cd.

## 1. Introduction

With the development of the mining and chemical industries, the release of cadmium (Cd) into the environment is accelerating. For this reason, Cd pollution has become a global environmental concern [1,2,3]. Cadmium, a nonessential element, accumulates throughout the food chain and greatly threatens human health. Cd compounds are highly toxic, highly mobile, and not biodegradable. In humans, long-term intake of rice with Cd content over 0.2 mg·kg^−1^ may cause irreversible liver dysfunction, cartilage disease, and kidney damage [4]. Cd in crops inhibits plant respiration, photosynthesis, mineral uptake, and transpiration, significantly reducing crop yield [5] and affecting food quality [6]. Due to the rapid industrialization in China during the last 40 years, soil pollution with Cd significantly increased throughout China [7,8]. It was reported that the content of Cd in about 7% of cultivated soil in China is higher than the national farmland soil environmental quality standard [9]. Therefore, it is important to control the uptake of Cd by agricultural crops as well as reduce Cd bioavailability in soil.

For this purpose, a variety of strategies were trialed, including physical treatment, chemical treatment, microbial remediation, and phytoremediation, among others [10]. Among these strategies, in situ stabilization technology with nontoxic soil modifiers effectively reduced Cd bioavailability and subsequent accumulation in crops. In this process, soluble Cd is transformed into a stable phase through adsorption, precipitation, and complexation [11]. Many soil modifiers, such as lime, iron-based materials, biochar, hydroxyapatite, zeolite, and crop straws, were added during the remediation of soil polluted with heavy metals [12]. Biochar represents a low-cost, environmentally friendly soil modifier widely used in the remediation of soil polluted with different heavy metals [13,14]. Previous studies indicated that Cd in soil can be fixed by using biochar [15,16]. Xu et al. reported that in the presence of nut-shell biochar, the bioavailability of Cd in soil was reduced by 49% [17]. Bashir et al. reported that the application of bagasse-based biochar in soil contaminated with cadmium and chromium reduced the availability of both heavy metals by 85% and 63%, respectively [18]. However, in real applications, biochar did not always show satisfactory Cd immobilization performance [19]. In addition, biochar materials alone show limited effects on Cd stabilization and are largely affected by soil properties such as pH, redox potential, organic matter content, and microbial activity [7,20,21]. Therefore, it is necessary to modify these materials in order to improve remediation performance.

Magnetic property modifications, acid-base property modifications, modification with minerals, and functional group grafting are some techniques that can be used for biochar modification. These techniques effectively improve biochar adsorption capacity for heavy metals [22,23,24,25,26]. Chen et al. proved that sulfur-modified wheat straw biochar reduced the available Cd content in soil by 22.72–27.90%, and also increased soil pH. The primary mechanism for Cd immobilization was the formation of CdS and CdHS^+^, as well as interaction with organic sulfides [27]. Xiao et al. used straw biochar and nitrogen fertilizer for the in situ fixation of Cd in soil, reporting that an appropriate amount of nitrogen significantly reduced Cd bioavailability and increased biomass in plant stems [28]. Zhu et al. confirmed that amine-modified biochar displayed high Cd adsorption capacity, up to 23.54 mg·kg^−1^ [29]. Sulfur- and nitrogen-modified biochar are the most common technologies. The introduction of sulfur alone or nitrogen alone effectively improves the immobilization capacity of biochar for cadmium ions in soil. However, research on the simultaneous introduction of both nitrogen and sulfur for the remediation of Cd-polluted soil is rarely reported. As a cheap and widely used surface modifier, thiourea could increase the bonding of amine and mercaptan functional groups and improve the adsorption capacity of heavy metals. In our previous work, we investigated the removal efficiency of Cd in aqueous solutions by using thiourea-modified biochar (TPBC600) [30]. However, cadmium adsorption in aqueous solutions was significantly different from that in soil colloids. Thus, the immobilization capacity of TPBC600 for cadmium ions present in soil still needs to be studied. There is also a lack of measured data on the effects of biochar on the physicochemical properties and cadmium immobilization of soil. Therefore, in the present study, we propose: (1) To evaluate the influence of modified biochar on the physical and chemical properties of Cd-contaminated soil; (2) to determine the influence of poplar-bark biochar (PBC600) and TPBC600 on Cd availability and Cd speciation in contaminated soil; and (3) to reveal the key factors affecting Cd immobilization in the soil system using Pearson correlation analysis, multivariate linear regression analysis, and principal component analysis. This information could be helpful in improving the remediation performance of biochar in Cd-polluted soil.

## 2. Materials and Methods

### 2.1. Contaminated Soil Sampling and Biochar Preparation

Topsoil samples were obtained at a depth between 0 and 20 cm in the National Reclamation Demonstration Region in Liuxin Town (34°23′43″ N and 117°07′29″ E), Tongshan County, Xuzhou City, Jiangsu Province, China. The area has a temperate, semihumid, continental, monsoon climate with an average annual precipitation of 8009−30 mm. The annual average temperate is 14 °C, and the frost-free period is 209 days [31]. Before pot experiments and physical and chemical analyses, the soil samples were dried in air, homogenized, and sieved to remove impurities. Analyses were performed according to the soil agrochemical analysis methodology of China [32]. Soil pH and electrical conductivity (EC) were determined after dispersing soil in water at a ratio of 1:2.5 (*w*/*v*) with a pH meter. Soil organic matter (SOM) was determined using the potassium dichromate methodology. Total nitrogen content (TN) was determined using the Kelvin distillation and titration method [33]. The ammonium nitrogen (${\mathrm{NH}}_{4}^{+}-N$) content was quantified using spectrophotometry after soil extraction with potassium chloride. The nitrate nitrogen (${\mathrm{NO}}_{3}^{-}-N$) content was determined using spectrophotometry after soil extraction with calcium chloride. In addition, available phosphorus content (AP) was determined using molybdenum–antimony–scandium colorimetry after soil extraction with ammonium bicarbonate. Also, flame photometry was applied to quantify available potassium (AK) in ammonium acetate soil extracts [34]. Turbidimetry with barium sulphate was used to determine available sulfur (AS) after soil extraction with calcium chloride. Each sample was determined thrice. The relative standard deviations (RSD) were below 10%. The primary physical and chemical properties are shown in Table 1.

In the present research, Italian poplar bark was used to prepare the biochar. For this purpose, the poplar bark was pyrolyzed for 2 h at 600 °C under argon atmosphere. The resulting material was labeled as PBC600. In addition, the modified biochar (TPBC600) was prepared as reported by Zhu et al. [30]. Briefly, a mixture with a mass ratio of 1:1 of poplar bark and thiourea was thoroughly mixed and placed in a tubular furnace. The mixture was slowly pyrolyzed to 600 °C in argon and this temperature was maintained for 2 h. The resulting biochar was sieved with a 60 mm mesh sieve and stored for future use.

### 2.2. Experimental Design and Sample Collection

PBC600 and TPBC600 samples were added to 500 g of air-dried soil at mass ratios of 1%, 2%, 4%, and 8% (*w*/*w*), and the mixtures were labeled as D1, D2, D4, and D8, respectively. The soil and biochar were evenly mixed, placed in a polyethylene bottle, and sealed with a cover. Soil was sprayed with deionized water on a daily basis in order to maintain a moisture content of 70%. The soil was incubated at 25 °C for 30 days. Biochar-free soil was set as the control group (CK). Each treatment was set up with three replicates. Sampling was performed at days 10, 20, and 30, and portions of the samples were stored at 2–0 °C for the determination of ammonium nitrogen and nitrate nitrogen contents. The other portions were dried in air and sieved with a 0.15 mm pore-diameter sieve and stored for future use.

### 2.3. Measurements and Analysis

Procedures used for the physical and chemical characterization of PBC600 and TPBC600 were as reported by Zhu et al. [30]. Total Cd content was determined using inductively coupled plasma mass spectrometry (ICP-MS) after soil digestion with HNO_3_-HF [35]. In addition, the four-step successive extraction method proposed by the European Community Bureau of Reference (BCR) provided information about the chemical forms of Cd present in the soil, i.e., Cd soluble in weak acids, reducible, oxidizable, and residual Cd [21]. The toxicity characteristic leaching method (TCLP) was used to determine the content of Cd in a leachate [7]. Analyses were performed in triplicate and the relative standard deviations (RSD) were smaller than 10%.

### 2.4. Data Analysis

Statistical analysis was performed using the SPSS 22.0 software. One-way ANOVA and Duncan test were selected to determine significant differences between groups under the probability condition of *p* < 0.05. Image and table data were processed using the Origin 2016 software. Pearson correlations were obtained using R. Multivariate linear regression (MLR) and principal component analysis (PCA) were applied to quantify the contributions of different factors on Cd immobilization in the soil. MLR was carried out with TCLPCd content as the dependent variable and the absolute principal component score as the independent variable, respectively.

## 3. Results

### 3.1. Effects of the Addition of Biochar on Soil Physicochemical Properties

Figure 1 shows that pH, EC, SOM, TN, ${\mathrm{NH}}_{4}^{+}-N$, ${\mathrm{NO}}_{3}^{-}-N$, AK, AP, and AS values significantly changed after the addition of biochar to the soil. Compared to the control group (CK), biochar increased soil pH (*p* < 0.05) (Figure 1a). In addition, after 10 days of incubation, the addition of 4% and 8% TPBC600 resulted in an increase of about 17% in soil pH compared to CK. Changes in EC were inversely proportional to changes in pH. In addition, the effect of biochar on soil pH and EC decreased over time. It was also observed that SOM, TN, AP, AK, and AS values improved after addition of biochar. The SOM values of all the TPBC600-treated groups were lower than those of the PBC600-treated groups. SOM reached a maximum value of 19.69 g·kg^−1^ on day 10 in the D8 group, which contained PBC600. Similar to other parameters, the effect of biochar on SOM declined over time (Figure 1b). PBC600 and TPBC600 displayed different effects on TN content (Figure 1c). Compared to CK, the application of PBC600 did not modify the TN value. However, TPBC600 significantly increased TN (*p* < 0.05). Also, the application of TPBC600 increased ${\mathrm{NH}}_{4}^{+}-N$ and ${\mathrm{NO}}_{3}^{-}-N$ contents in soil to a larger extent compared to PBC600. Nevertheless, this positive effect decreased with increasing incubation time (Figure 1d,e). Results for AP indicated that the application of biochar effectively improved the availability of soil phosphorus. AP values increased between day 10 and 20, where values were constant after these times (Figure 1g). AK values indicated that, during the 30 days of incubation, the content of available potassium in the soil was slightly higher than that of CK. In addition, available potassium in the soil treated with TPBC600 was higher than that treated with PBC600 (Figure 1f). Similarly, available sulfur in the TPBC600 treatment was higher than that in the PBC600-added soil (*p* < 0.05). It was noteworthy that the content of available sulfur in soil drastically decreased as incubation time increased, which may be explained by the decomposition of biochar over time (Figure 1h).


### 3.2. Effect of Biochar on Cd Availability in Soil

Data on available Cd in soil can be used to evaluate potential Cd toxicity in natural environments [36]. The levels of TCLPCd in the different treatments used in the present research are shown in Figure 2. The data indicated that Cd was immobilized by the PBC600 and TPBC600 treatments. In addition, immobilization was higher in the TPBC600 treatment compared to PBC600. With the increase in biochar content, the available Cd in the soil significantly decreased (*p* < 0.05). As incubation time increased, Cd immobilization in the treated soils also increased. After 30 days of incubation with PBC600, the available Cd in the soils with added D1, D2, D4, and D8 were reduced by 36.26%, 47.03%, 49.01%, and 55.38%, respectively, compared to the CK group. In similar conditions, values for available Cd content in TPBC600-treated soils were reduced by 79.56%, 79.78%, 80.22%, and 82.86%, in the D1, D2, D4, and D8 groups, respectively. The adsorption capacity of TPBC600 was nearly five times higher than biochar after pyrolysis with thiourea impregnation, with the reason for this difference possibly being the different activation of raw biochar in one-step copyrolysis [37]. Moreover, different feedstocks and modifiers changed the surface characteristics, minerals, and functional groups of biochar. Thus, environmental risks caused by available Cd in soil were significantly reduced after treatment with these biochar materials.

### 3.3. Effect of Biochar on Cd Speciation in Soil

After incubation, the chemical forms of Cd in the soils were determined using the BCR continuous extraction procedures. As shown in Figure 3, concentrations of residual Cd in soil samples containing 1%, 2%, 4%, and 8% TPBC600 increased after 30 days by 29.71%, 31.54%, 30.08%, and 32.57%, respectively, compared with the CK treatment. In addition, concentrations of acid-soluble Cd decreased by 9.11%, 13.20%, 14.33%, and 15.59%, respectively. It was also observed that, with increased TPBC600 content, residual Cd in the soils significantly increased and acid-soluble Cd significantly decreased. Also, as incubation time increased, these trends were more evident. On the other hand, in the soil samples containing 4% and 8% PBC600, the content of residual Cd increased by 1.16% and 8.48%, correspondingly. In this case, as incubation time increased, the concentration of acid-soluble Cd slightly decreased, indicating that the TPBC600 treatment reduced Cd bioavailability to a higher extent compared to PBC600. It was also observed that, even after 10 days of incubation, soils treated with PBC600 presented a higher content of residual Cd compared to controls. Comparing Cd distribution in PBC600 and TPBC600, the results indicated that the acid-soluble and reducible Cd fractions could be converted into oxidizable and residual Cd fractions. TPBC600 was more effective than PBC600 with regard to this process.

## 4. Discussion

### 4.1. Effects of the Addition of Biochar on Soil Nutrients and Cd Remediation

In the present study, the effectiveness of biochar in improving soil acidity, enhancing the contents of nutrients in soil, and reducing Cd bioavailability was investigated. Changes in soil properties after the application of biochar may be either caused by the biochar itself or by the interactions between physical and chemical properties. The increase in soil pH may be related to the biochar pH. After application, the basic ions present in the biochar, including Ca^2+^, Mg^2+^, and K^+^, were released in the form of carbonates and oxides, and these cations were exchanged with acidic ions. As a result, pH increased [38]. In addition, the negatively charged carboxyl and hydroxyl groups, as well as phenolic groups present on the biochar surface, interacted with acidic ions, reducing H^+^ concentrations [39]. Furthermore, soil pH was the main factor affecting phosphorus immobilization in soil [40,41]. The alkalinity and content of available phosphorus in soil increased after biochar treatment. The high specific surface area and abundant functional groups of biochar promoted the presence of adsorption sites for phosphate ions [42]. Also, EC differences may be attributed to dissimilar abilities of both biochar materials to improve the content of water-soluble ions in soil [43]. Therefore, compared to PBC600, TPBC600 displayed a weaker capacity for ion retention. The organic matter content in soil is a key factor that affects soil fertility and the combination of heavy metals present in soil matter. In this research, changes in organic matter contents in soil were consistent with organic carbon contents in biochar. This indicated that organic matter content in soil was positively correlated with organic carbon content in biochar. The increase in organic carbon content in the soils treated with biochar may be the result of (a) an increase in organic matter decomposition, or (b) increased absorption of organic molecules in soil particles [17]. Modification with thiourea decreased the content of volatiles and soluble organic carbon, resulting in reduced mineralization and decomposition processes in TPBC600. Similar to the trends observed for pH and SOM, the increase in TN may be related to the nitrogen content of biochar. The application of biochar increased the content of ${\mathrm{NH}}_{4}^{+}-N$ and ${\mathrm{NO}}_{3}^{-}-N$, with the increase in ${\mathrm{NH}}_{4}^{+}-N$ possibly being caused by the decomposition of organic matter in the biochar, and the increase in ${\mathrm{NO}}_{3}^{-}-N$ levels by a more severe nitrogen mineralization [44]. This indicates that biochar can not only provide mineralized nitrogen to soil, but also promotes the nitrogen cycle in soil. The available potassium in the biochar can be directly utilized by soil. Previous reports indicated that the direct contribution of biochar to available potassium in soil was insignificant. In addition, increased levels of available potassium may be the result of soil potassium immobilization and activation by biochar [45,46]. In this work, we observed that the content of available potassium (AK) exhibited an interesting changing trend during the incubation period. In the early stage of incubation, the direct contribution of available potassium present in the biochar to available potassium in soil was dominant, and this contribution was gradually reduced as time increased. In addition, potassium immobilization and activation by the biochar played a dominant role during the late incubation stage.

Cd inhibition and immobilization in soil may be partially attributed to the biochar. After soil modification, the specific surface areas and porous structures of both biochar materials increased, as did the types of negatively charged functional groups on the surfaces. These functional groups are able to bond Cd through chelation and complexation interactions [47]. In addition, a lower C/N ratio effectively reduces the availability of Cd in soil [39]. Soil pH is an important parameter affecting the species and mobility of heavy metals in contaminated soil. The increased Cd immobilization and decreased Cd mobility after biochar application may be attributed to the mechanisms of complexation, adsorption, and/or precipitation. Also, the increase in both soil pH and buffer capacity may further improve Cd uptake [15]. It was reported that increased pH increases the density of negative charges in soil, as well as the density of negative charges in biochar particles causing Cd immobilization [6,48]. In addition, the ash in biochar may induce the coprecipitation of Cd with anions and cations, such as Ca^2+^, Mg^2+^, and (PO_4_)^3-^ [7]. After the modification, the number of cationic exchange sites increased due to the increase of sulfur- and nitrogen-containing groups. The latter, especially amines (-NH_2_), may promote Cd immobilization through the formation of strong covalent bonds [49]. The formation of cadmium sulfide may also reduce the content of available Cd in soil. Our results showed that the applications of PBC600 and TPBC600 may significantly reduce the environmental risk of Cd present in soil.

### 4.2. Fundamental Factors Affecting the Availability of Cd in Soil

Pearson correlation analysis was used to evaluate the relationships between physical and chemical properties of the soils, incubation days, biochar dosage, and available Cd content in the soils (Figure 4). Data indicated that, in the group treated with PBC600, soil pH, AP, and TN content were negatively correlated with available cadmium (TCLPCd) (*p* < 0.001). In the TPBC600 treatments, soil pH, EC, TN, and AP were negatively correlated with TCLPCd. Furthermore, nitrate nitrogen content and AK were negatively correlated with TCLPCd. Pearson correlation analysis also showed that the available cadmium in soil was synergistically affected by multiple factors. Among them, pH and AP displayed the greatest influence on concentrations of available cadmium in soil. According to the data, improvement in soil physical and chemical properties was higher when TPBC600 was applied compared to PBC600. This occurred because of the unique adsorption performance, structure, and composition of the resulting biochar modification. This uniqueness may be the main reason for the difference in the physical and chemical properties of soil samples treated by those two biochar materials. The multivariate linear regression equations correlating soil physical and chemical properties with the content of available cadmium in soil are shown in Equations (1) and (2). pH and AP displayed the greatest influence on Cd availability in soil. The results of Pearson correlation analysis and multivariate linear regression equations showed that soil pH and AP content may be the main factors restricting the reduction in Cd availability. The application of TPBC600 significantly increased soil pH. Moreover, with increasing pH, the number of H^+^ ions present in the soil competing for Cd^2+^ adsorption decreased, and soil adsorption capacity for Cd^2+^ increased [50]. In addition, the increase in pH promoted the hydrolysis of more Cd^2+^ ions, and these ions were easier to adsorb on biochar and soil [51]. In addition, when the soil pH was high enough, Cd^2+^ in soil reacted with OH^-^, forming a hydroxide and precipitating [52].
(1)$$\begin{array}{c} {\mathrm{TCLPCd}}_{\left(\mathrm{PBC}600\right)}\;=\;12.631-0.364\mathrm{AP}-1.431\mathrm{pH}+0.057\mathrm{AK}-0.048\mathrm{AS}+0.056\mathrm{SOM}-0.026{\mathrm{NO}}_{3}^{-}-N\;–\\ 0.099\mathrm{EC}\;+\;0.003{\mathrm{NH}}_{4}^{+}N\;-\;0.01\mathrm{TN} \end{array}$$
(2)$$\begin{array}{c} {\mathrm{TCLPCd}}_{\left(\mathrm{TPBC}600\right)}\;=\;19.70-2.326\mathrm{pH}-0.14\mathrm{AP}-0.04\mathrm{SOM}\;+\;0.346\mathrm{EC}\;+\;0.066{\mathrm{NO}}_{3}^{-}N\;+\;0.025\mathrm{AK}\;–\\ 0.002{\mathrm{NH}}_{4}^{+}N-0.013\mathrm{AS}\;+\;0.013\mathrm{TN} \end{array}$$

Even when soil pH and AP were the most influential parameters on Cd availability in soil, other physical and chemical properties also affected this to a certain extent. In general terms, the effects of SOM on the mobility of heavy metals depend on composition and content. The formation of highly soluble organic complexes from small organic acids favors heavy metal mobility. On the other hand, low-solubility, metal-macromolecular, organic chelates immobilize heavy metals. Thus, the influence of SOM on the availability of heavy metals is dependent on both effects [53,54]. Also, nitrogen content indirectly affected the concentrations of heavy metals in soil. This occurred because of changes in soil pH and other parameters. Ammonium generated from the hydrolysis of nitrogen participated in the nitrification process under aerobic conditions. As a consequence, soil pH and heavy metal availability were affected [44].

In order to understand the differences in cadmium availability in the different treatments used in the present research, PCA of the physical and chemical properties of the soils and the concentrations of available cadmium on different incubation days were separately conducted. The first two principal components were extracted from each set of data (Figure 5). Figure 5 shows that the relationship between the physical and chemical properties of the soils and the available cadmium were not singular and regular, but multivariate and changeable. On different incubation days, the relationships between the physical and chemical properties of the soils and the Cd availability were different. After 10 days of incubation, the results from the CK group, the D8 group treated with PBC600, and the D1 group treated with TPBC600 were similar. This indicated that the effects of the different treatments on reducing TCLPCd were also similar. TN, AS, and pH were the main factors that affected the content of available Cd in the soils. These relationships were indicated by the cosine values of the angles between TCLPCd and the physical and chemical properties of the soils. Therefore, it can be assumed that, at the initial stage of incubation, the direct influence of the principal properties of biochar on soil, resulting in changes in the physical and chemical properties of the soils, should be thought of as the most important factors determining the difference in available cadmium content in the soils treated by those two biochar materials. As shown in Figure 5b,c, the distances between the CK and both biochar treatments were large, indicating that the reduction in available cadmium in the CK group was extremely different to that observed in the biochar treatments. Results for the other PBC600 and TPBC600 treatments were relatively concentrated, as shown in the graph, and the trend of concentration was more evident as incubation time increased. However, the data points corresponding to the treatments with those two biochar materials were located in different quadrants. In summary, for the same biochar treatment, the variation patterns regarding the content of available cadmium were similar; however, these treatments displayed slightly different effects on changing ability for Cd availability in the soil media.

## 5. Conclusions

Cd pollution in soils is a major environmental issue related to the development of mining and chemical industries. In the present study, pot experiments were conducted to determine the effects of poplar-bark biochar (PBC600) and thiourea-modified poplar bark biochar (TPBC600) on (a) the physical and chemical properties of soil and (b) Cd speciation and availability. The results showed that the addition of biochar significantly affected the physicochemical properties of soil, and enhanced soil pH, EC, SOM, TN, ${\mathrm{NH}}_{4}^{+}-N$, ${\mathrm{NO}}_{3}^{-}-N$, AK, AP, and AS. The addition of biochar was able to transform acid-soluble and reducible Cd fraction into oxidizable and residual Cd fractions, thus reducing the bioavailability of Cd in soil. This was mainly attributed to changes in soil pH and the content of available phosphorus. According to our results, TPBC600 was more effective than PBC600 in the remediation of Cd-contaminated soil, which may be explained by the addition of nitrogen and sulfur for the improvement in adsorption capacity. Therefore, TPBC600 is believed to be an effective modifier for the remediation of Cd-contaminated soil.

## Figures and Tables

**Figure 1 ijerph-17-07654-f001:**
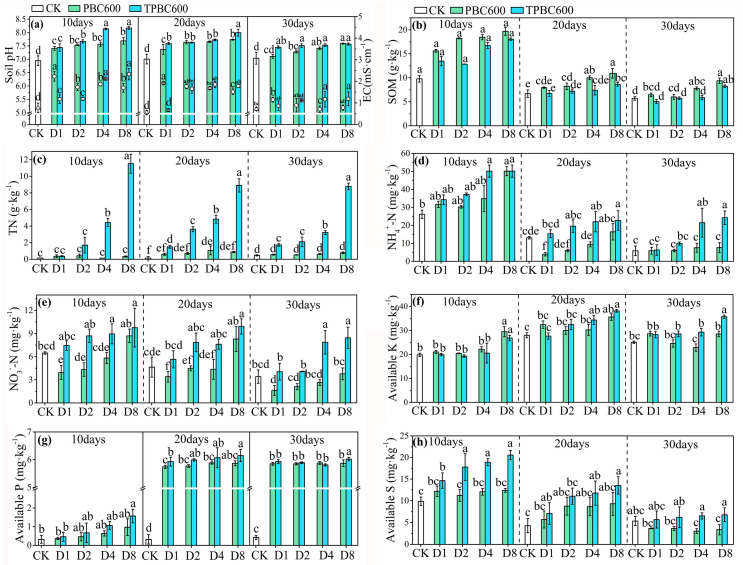
Effects of poplar-bark biochar (PBC600) and thiourea-modified poplar-bark biochar (TPBC600) treatments on pH, electrical conductivity (EC) (**a**), organic matter (SOM) (**b**), total nitrogen (TN) (**c**), ammonium nitrogen (${\mathrm{NH}}_{4}^{+}-N$) (**d**), nitrate nitrogen (${\mathrm{NO}}_{3}^{-}-N$) (**e**), available potassium (AK) (**f**), available phosphorus (AP) (**g**), and available sulfur (AS) (**h**) in soil. The triangles, circles, and squares in Figure 1a represent the EC values of CK, PBC600, and TPBC600, respectively. The data and error bars represent the mean values and standard deviations. Different letters indicate that the average values under different sampling conditions were significantly different (*p* < 0.05).

**Figure 2 ijerph-17-07654-f002:**
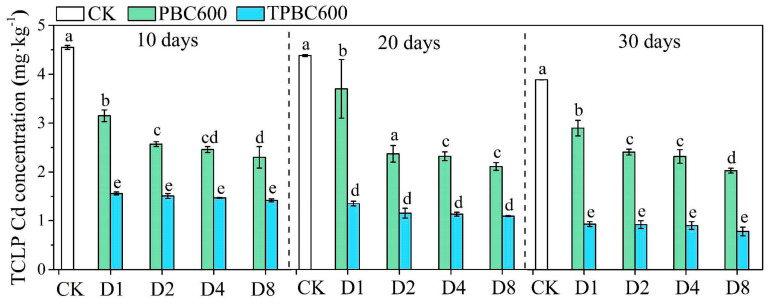
Effects of PBC600 and TPBC600 on cadmium availability in soil. Different letters indicate that the average values under different sampling conditions were significantly different (*p* < 0.05).

**Figure 3 ijerph-17-07654-f003:**
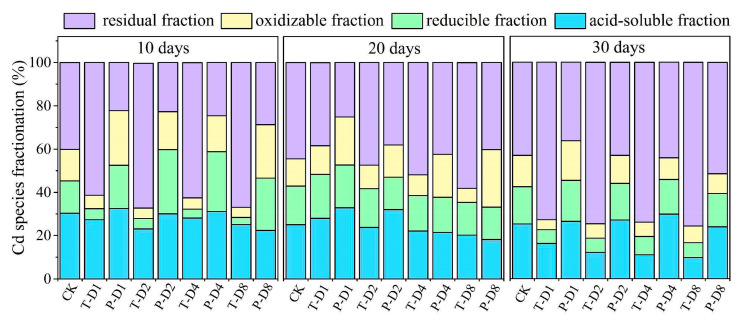
Influence of PBC600 and TPBC600 on distribution of Cd species. Note: T stands for TPBC600; *p* stands for PBC600.

**Figure 4 ijerph-17-07654-f004:**
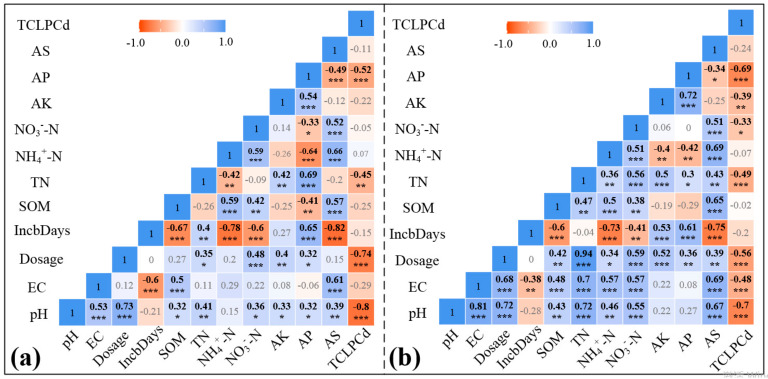
Pearson correlation matrixes for the relationships between physical and chemical properties of the soils, incubation days, biochar dosages, and availability of Cd in the soils. (**a**) PBC600; (**b**) TPBC600. * indicates that *p* < 0.05 (significant); ** indicates that *p* < 0.01 (very significant); *** indicates that *p* < 0.001 (extremely significant).

**Figure 5 ijerph-17-07654-f005:**
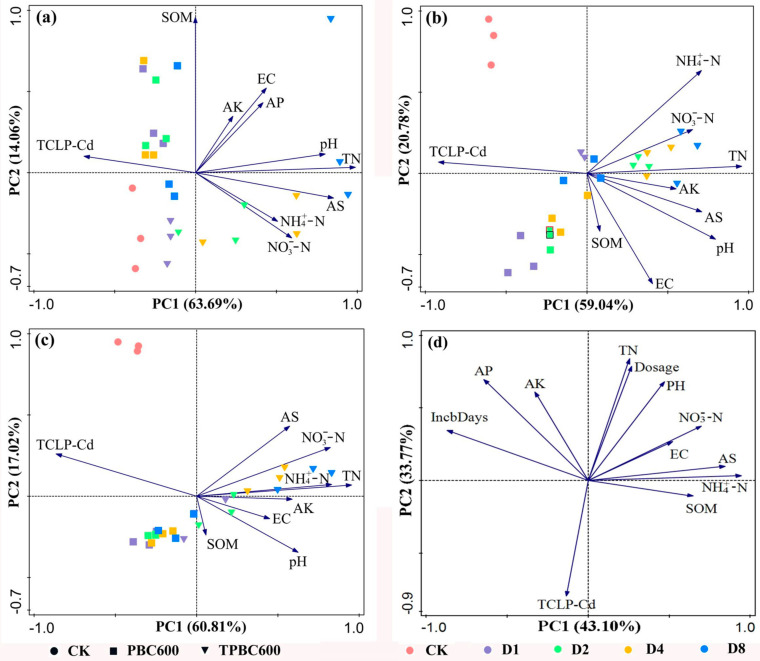
Principal component analysis results of the physical and chemical properties and available Cd content in soil. (**a**) Ten days of incubation, (**b**) 20 days of incubation, (**c**) 30 days of incubation, (**d**) physical and chemical properties of the soils, incubation time, biochar dosage, and available Cd content in the soils.

**Table 1 ijerph-17-07654-t001:** Physical and chemical properties of soil and biochar.

Properties	Soil	PBC600	TPBC600
pH	6.86 ± 0.45 c	9.92 ± 0.21 b	10.85 ± 1.00 a
Ash (%)	-	41.68 ± 0.52 b	47.71 ± 0.29 a
Average pore diameter (nm)	-	1.94 ± 0.03 b	3.42 ± 0.01 a
Surface area (m^2^·g^−1^)	-	2.77 ± 0.18 b	5.70 ± 0.35 a
EC (mS·cm^−3^)	0.20 ± 0.25 a	0.38 ± 0.48 a	0.43 ± 0.17 a
SOM (g·kg^−1^)	10.38 ± 0.57 c	29.49 ± 0.33 a	17.65 ± 0.45 b
TN (g·kg^−1^)	0.21 ± 0.02 c	1.23 ± 0.06 b	17.32 ± 0.06 a
${\mathrm{NH}}_{4}^{+}-N$ (mg·kg^−1^)	1.98 ± 0.75 c	20.42 ± 0.40 b	40.72 ± 0.51 a
${\mathrm{NO}}_{3}^{-}-N$ (mg·kg^−1^)	1.38 ± 1.17 c	2.58 ± 0.18 b	15.03 ± 0.32 a
AP (mg·kg^−1^)	0.86 ± 0.06 c	9.50 ± 0.00 a	8.81 ± 0.16 b
AK (mg·kg^−1^)	22.30 ± 0.20 b	36.64 ± 0.95 a	14.03 ± 0.27 c
AS (mg·kg^−1^)	11.77 ± 0.77 b	6.64 ± 1.40 c	25.48 ± 1.05 a
Cd (mg·kg^−1^)	9.97 ± 0.01 a	0.04 ± 0.01 c	0.06 ± 0.01 b

Note: Data are mean ± standard deviation. EC, electrical conductivity; SOM, soil organic matter; TN, Total nitrogen; ${\mathrm{NH}}_{4}^{+}-N$, ammonical nitrogen; ${\mathrm{NO}}_{3}^{-}-N$, nitrate nitrogen; AP, available phosphorus; AK, available potassium; AS, available sulfur. The same below. Data were compared among treatments (Soil, PBC600 and TPBC600) using Tukey honestly significant difference test following ANOVA. Numbers with the same letter are not significantly different at *p* = 0.05. Numbers with the different letter are significantly different at *p* = 0.05.

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
