# Peer review of "Remediation of Soil Polluted with Cd in a Postmining Area Using Thiourea-Modified Biochar"

_ijerph, 2020, doi:10.3390/ijerph17207654_

Round 1

Reviewer 1 Report

The authors have assessed the effect of thiourea-modified biochar to remediate Cd-polluted soil. Overall, the paper is interesting, well-written, and reports relevant findings. Yet, there are a number of issues that need to be addressed before publication. Please find below a list of specific comments.

- Language: There are a number of grammar and syntax mistakes across the manuscript. Therefore the language should be carefully revised, if possible by a native English speaker. Pay particular attention to the abstract, as the first few sentences are especially poorly written.

- Although the authors mention that the soil is from a mining area, the information about the soil/area is scarce. Please provide further details.

- Introduction: The authors do not explain the reason for the selection of thiourea for biochar modification.

- Table 1: Average pore diameter of PBC600 and TPBC600?

- Table 1: The table would benefit from a statistical analysis (t-test) between PBC600 and TPBC600 to check for significant differences on the selected properties.

- Since this is not the first paper that employs thiourea-modified biochar to remediate Cd-polluted soil, the authors should thoroughly compare and discuss their results with: Leila Gholami, Ghasem Rahimi & Abolfazl Khademi Jolgeh Nezhad (2020) Effect of thiourea-modified biochar on adsorption and fractionation of cadmium and lead in contaminated acidic soil, International Journal of Phytoremediation, 22:5, 468-481, DOI: 10.1080/15226514.2019.1678108

- It would also be interesting to discuss the implications of the findings for phytoremediation of Cd-polluted soils, particularly in mining areas.

Reviewer 2 Report

This study appears well performed and well presented. I only have minor comments.

Line 13: "presence in" instead of "present".

Line 23: The improvement was diminished? It is unclear if this means that the effect disappeared? Or that PBC600 was almost as effective but the effect was slower (and the difference in effect was diminished)?

Line 110: Add a unit for the hole size in the 60-mesh sieve.

Table 1: All the abbreviations should be explained in the text also. EC should also be explained in the table legend. It should furthermore be written out what the elements P, K, and S are.

Line 139: "..obtained using R".

Line 139: Complete the sentence on MLR.

Line 142-143: "..as the independent variable, the results of absoulute principal components." is not a full sentence and has to be completed.

Figure 3: "acid-soluble" instead of "acid-souble".

Line 334: I think ", which may threathen human health," is not necessary here and can be deleted.

Line 339: What is meant by improved? Levels were reduced?

Round 2

Reviewer 1 Report

The authors have satisfactorily addressed my remarks. I consider that the manuscript has been significantly improved and is now suitable for publication.